# Adapting clinical chemistry plasma as a source for liquid biopsies

Spencer C Ding[1], Jingru Yu[1], Tiepeng Liao[1], Lauren Ahmann[2], Yvette Yao[3], Chandler Ho[4], Linlin Wang[5], Benjamin A Pinsky[1,6], Wei Gu[1]*

[1]Department of Pathology, School of Medicine, Stanford University, Stanford, United States; [2]Department of Molecular and Medical Genetics, Oregon Health & Science University, Portland, United States; [3]University of British Columbia, Vancouver, Canada; [4]Clinical Laboratories, Stanford Health Care, Stanford, United States; [5]Department of Laboratory Medicine, School of Medicine, University of California, San Francisco, San Francisco, United States; [6]Department of Medicine, Division of Infectious Diseases and Geographic Medicine, School of Medicine, Stanford University, Stanford, United States

## eLife Assessment

This **important** work provides a new method to extract cfDNA from residual plasma from heparin separators for molecular testing. The evidence supporting the authors' claims is **convincing**, although some further metrics should also be evaluated. This finding will be interesting to people working in epigenomics and infectious disease diagnostics.

*For correspondence:
weigu@stanford.edu

**Abstract** Circulating cell-free DNA (cfDNA) is valuable for molecular testing, but typically requires specialized collection tubes or immediate processing. We investigated whether residual plasma from heparin separators, routinely used in clinical chemistry, could serve as an accessible and underused source for cfDNA. We analyzed matched plasma samples from healthy volunteers in two experiments: an immediate-processing comparison across EDTA, Streck, and heparin separator tubes (n=5), and a clinical-handling simulation comparing EDTA and heparin separator tubes under delayed processing at room temperature or 4°C (n=6). We also analyzed matched plasma samples from viral PCR-positive patients in a hospital cohort (n=38). Whole-genome sequencing and enriched methylation sequencing were performed to assess concordance across metagenomics, copy number, methylation, and fragmentomic features. Under immediate processing, heparin separator plasma showed high concordance with EDTA and Streck plasma for methylation patterns (Spearman's $\rho$ =0.65–0.70) and fragmentation features. In the Hospital Cohort, heparin separator plasma showed strong concordance with matched EDTA plasma for viral detection (Spearman's $\rho$ =0.95), copy number alteration profiling (Spearman's $\rho$ =0.72–0.96), and methylation patterns (Spearman's $\rho$ =0.50–0.83). These findings support the feasibility of using refrigerated, promptly processed residual plasma from routine clinical chemistry as a supplementary source for cfDNA biobanking and molecular analyses.

## Introduction

Plasma cell-free DNA (cfDNA) is widely used in clinical testing (*Loy et al., 2024*), including noninvasive prenatal testing (*Lo et al., 1997*; *Fan et al., 2008*; *Lo et al., 2010*; *Fan et al., 2012*), cancer testing (*Chan et al., 2017*; *Chung et al., 2024*; *Heitzer et al., 2015*), infectious disease testing (*Gu et al., 2019*; *Blauwkamp et al., 2019*), and organ transplantation monitoring (*De Vlaminck et al., 2014*;

*Bloom et al., 2017*). In cancer testing, cfDNA is actively studied for the detection of residual cancer disease (*Lennon et al., 2020*) and early detection (*Shen et al., 2018*; *Jamshidi et al., 2022*).

Reliable cfDNA analysis typically depends on specialized preservative tubes (e.g. Streck) or rapid processing of EDTA tubes (*Meddeb et al., 2019*). However, broader implementation is limited by the requirement for specific tubes or on-call personnel. In contrast, heparin separators are now ubiquitous in daily blood draws at many hospitals for routine chemistry testing, such as the basic metabolic panel. At Stanford Health Care alone, over 900,000 such tubes are processed annually. After testing, there is typically a per-tube surplus of 0.5–2 mL of plasma with cfDNA, which is discarded.

Despite its availability and appeal as a residual sample source, heparinized plasma has historically been avoided for molecular testing because (i) heparin is a well-established inhibitor of PCR, discouraging its use in PCR-based assays (*Beutler et al., 1990*; *Yokota et al., 1999*; *Jung et al., 1997*), (ii) genomic DNA (gDNA) leakage, and (iii) cfDNA degradation. Prior studies comparing heparinized and EDTA plasma samples have reported mixed results and often do not distinguish between gDNA contamination and cfDNA degradation in PCR measurements. *Barra et al., 2025* and *Gerber et al., 2020* reported elevated cfDNA levels in heparin tubes compared to specialized tubes such as PAXgene, Roche cfDNA, and Streck, even at time point 0. In contrast, Lo et al. and *Hebels et al., 2013* reported comparable cfDNA yield and integrity between EDTA and heparin plasma when samples were processed promptly (*Lam et al., 2004*) or stored at –80°C within 8 hr (*Hebels et al., 2013*). Using ddPCR, *van Ginkel et al., 2017* reported a lower cfDNA concentration in heparin samples compared with EDTA and serum samples. Notably, most existing studies relied on PCR-based assays for overall DNA quantification rather than quantification of cfDNA sequences detectable by NGS. Tracking sequenceable cfDNA directly provides measurements required for most cfDNA testing.

In the current work, we focus on heparin separator tubes, which differ from plain heparin tubes in that they contain a gel barrier that physically separates plasma from cellular components after centrifugation. Common chemistry tests are typically performed on-site at hospitals and affiliated clinics. These tubes are processed through high-throughput, automated clinical workflows designed to support rapid turnaround in a regulated environment. Moreover, typical NGS workflows include multiple purification steps that reduce heparin and gDNA contamination before the final PCR, and adapter-ligation workflows further reduce gDNA carryover. Together, these features motivate an evaluation of whether ubiquitous heparin separators are a fit for modern sequencing-based cfDNA analysis.

To assess the feasibility of using heparin separators for cfDNA testing, we conducted whole-genome sequencing (WGS) and the FLEXseq methylation assay (*Yu et al., 2024*) on paired samples collected from the same venipuncture in both heparin separators and gold-standard EDTA (and/or Streck) tubes. Our evaluation spanned multiple benchmarks relevant to a broad assortment of cfDNA tests, including metagenomics (*Gu et al., 2019*; *Gu et al., 2021*), methylation profiling (*Yu et al., 2024*; *Loyfer et al., 2023*), copy number alterations (CNAs) (*Lenaerts et al., 2019*), epigenetic tissue-of-origin mapping (*Loyfer et al., 2023*), and cfDNA fragmentomics (*Lo et al., 2021*).

## Results

### Heparin separators preserve cfDNA integrity under ideal conditions

To evaluate the potential of repurposing leftover plasma from heparin separators, we conducted a controlled Tube Comparison Study (n=5) in the Healthy Cohort using matched samples collected from the same blood draw in heparin separators and in gold-standard EDTA and Streck tubes (*Figure 1a*, *Supplementary file 1*).

The heparin separator samples showed coverage profiles comparable to the matched EDTA samples (*Figure 1—figure supplement 1*, Spearman's $\rho$ =1.00). Fragment size distributions were highly similar across all tube types, with a modal peak at ~166 bp, consistent with known cfDNA profiles (*Figure 1b*; *Lo et al., 2010*; *Fan et al., 2010*). End motif analysis of 256 4-mers showed comparable rankings (*Figure 1c*), suggesting that heparin separators do not perturb fragmentation patterns when processed immediately.

To assess the preservation of epigenomic features, we calculated methylation levels at individual CpG sites using the FLEXseq methylation assay's output. CpG sites with ≥30× coverage showed correlations between heparin separator and EDTA (Spearman's $\rho$ =0.70, *Figure 1d*, left), and

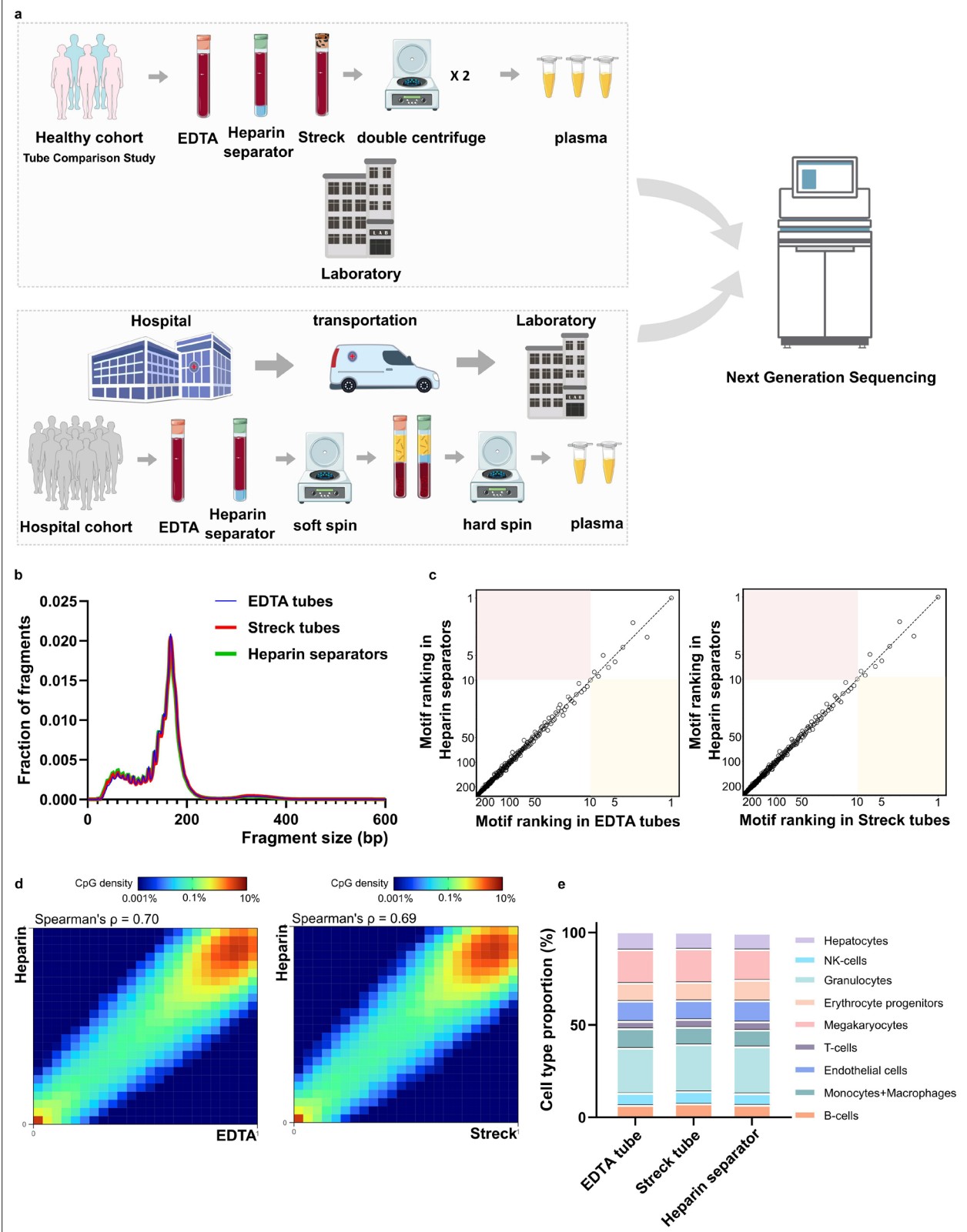

**Figure 1.** Overall schematic and Healthy Cohort's comparative performance. (**a**) Schematic illustration of the experimental design across two cohorts. The Healthy Cohort was a controlled experiment where plasma samples from healthy volunteer donors were collected freshly in EDTA, Streck, and heparin separators. The Hospital Cohort plasma samples were retrospectively collected in EDTA and heparin separators as part of patient care and leftover from routine clinical testing. The Hospital Cohort samples underwent an initial soft spin during the routine clinical workflow and then a hard spin

*Figure 1 continued on next page*

*Figure 1 continued*

after refrigerated storage. Processed samples were analyzed using whole-genome sequencing (WGS) and/or FLEXseq methylation testing. (**b**) Healthy Cohort fragment size distribution of cell-free DNA (cfDNA) collected across the three tubes. (**c**) Healthy Cohort comparison of end motif rankings in heparin separator samples versus EDTA or Streck tubes. (**d**) Healthy Cohort heatmap and correlation of methylation beta values in paired samples from a healthy donor (P140). Heparin separators (y-axis) are compared to EDTA and Streck tubes (x-axis). (**e**) Healthy Cohort estimation of cell-type proportions using methylation deconvolution for each collection tube.

The online version of this article includes the following figure supplement(s) for figure 1:

**Figure supplement 1.** Genome-wide coverage comparison between tube types.

**Figure supplement 2.** Healthy Cohort methylation correlation and cell-type deconvolution.

between heparin separator and Streck (Spearman's $\rho$ =0.69, *Figure 1d*, right). Tissue deconvolution revealed highly concordant cell-type proportions across all collection tube types (*Figure 1e*, p=0.86, PERMANOVA), with consistent results across all five cases (heparin separator versus EDTA: Spearman's $\rho$ =0.65–0.70; heparin separator versus Streck: Spearman's $\rho$ =0.65–0.69; *Figure 1—figure supplement 2*). These findings show that heparin separators preserve the biological status of methylation under ideal processing conditions.

To better reflect real-world handling procedures, we performed a controlled delay study in six healthy volunteers using paired EDTA and heparin separator tubes from the same blood draw (*Figure 2a*, *Supplementary file 2*). When compared with the corresponding control samples that were processed at time 0, minimal changes in fragment size distributions were observed across EDTA samples at all incubation times, at either room temperature or 4°C (*Figure 2b*). Heparin separator samples stored at 4°C at all incubation times also show similar profiles (*Figure 2b*). While in the heparin separator samples stored at room temperature showed a time-dependent shift toward shorter fragment size, which indicates increasing degradation (*Figure 2b*). As an extreme control, incubation at 37°C for 24 hr, as used in Barra et al., produced drastic fragmentation changes, reflecting a lot of degradation (*Figure 2b*). Together, these results indicate that temperature before the first spin is a key determinant of cfDNA integrity in heparin separator tubes.

## Hospital Cohort: residual plasma from heparin separators after clinical usage

To evaluate the feasibility of heparin separator tubes for cfDNA analysis, we identified 38 matched pairs of viral PCR-positive plasma samples collected in both heparin separator and EDTA tubes from the same blood draw. These cases were retrieved by querying the electronic medical record systems at Stanford (n=32) and UCSF (n=6) for samples that tested positive by the Clinical Virology Laboratories. All samples underwent cfDNA extraction, followed by WGS to evaluate viral load, CNAs, and fragmentation profiles, as well as methylation profiling via FLEXseq.

## Leftover plasma from the heparin separators enables viral load detection

We performed metagenomic sequencing to compare viral load measurements and assess concordance between plasma collected in EDTA and heparin separators (*Figure 3*). WGS achieved a median coverage of 4.33× (IQR: 2.89–4.88×). Viral read counts normalized to the number of reads aligned to the human genome were highly correlated between plasma collection tube types (Spearman's $\rho$ =0.95, p<0.001), with no significant difference observed (Wilcoxon signed-rank test, p=0.27). This high level of concordance was consistently observed across institutions and among distinct viral types.

## Leftover plasma from the heparin separator enables copy number profiling

CNAs are commonly observed in cancer and can be detected in cfDNA as a surrogate marker for tumor burden. To evaluate whether heparin separators preserve CNA signals, we compared the genome-wide copy number profiles of samples from heparin separators and EDTA tubes. CNA profiles were generated using a 500 kb bin size, with a median of 97.15 million reads per sample (range: 0.73–360.73 million). Representative genome-wide CNA plots from a matched pair (Patient 34) revealed similar patterns of chromosomal gains and losses (*Figure 4a*). Genome-wide CNA plots

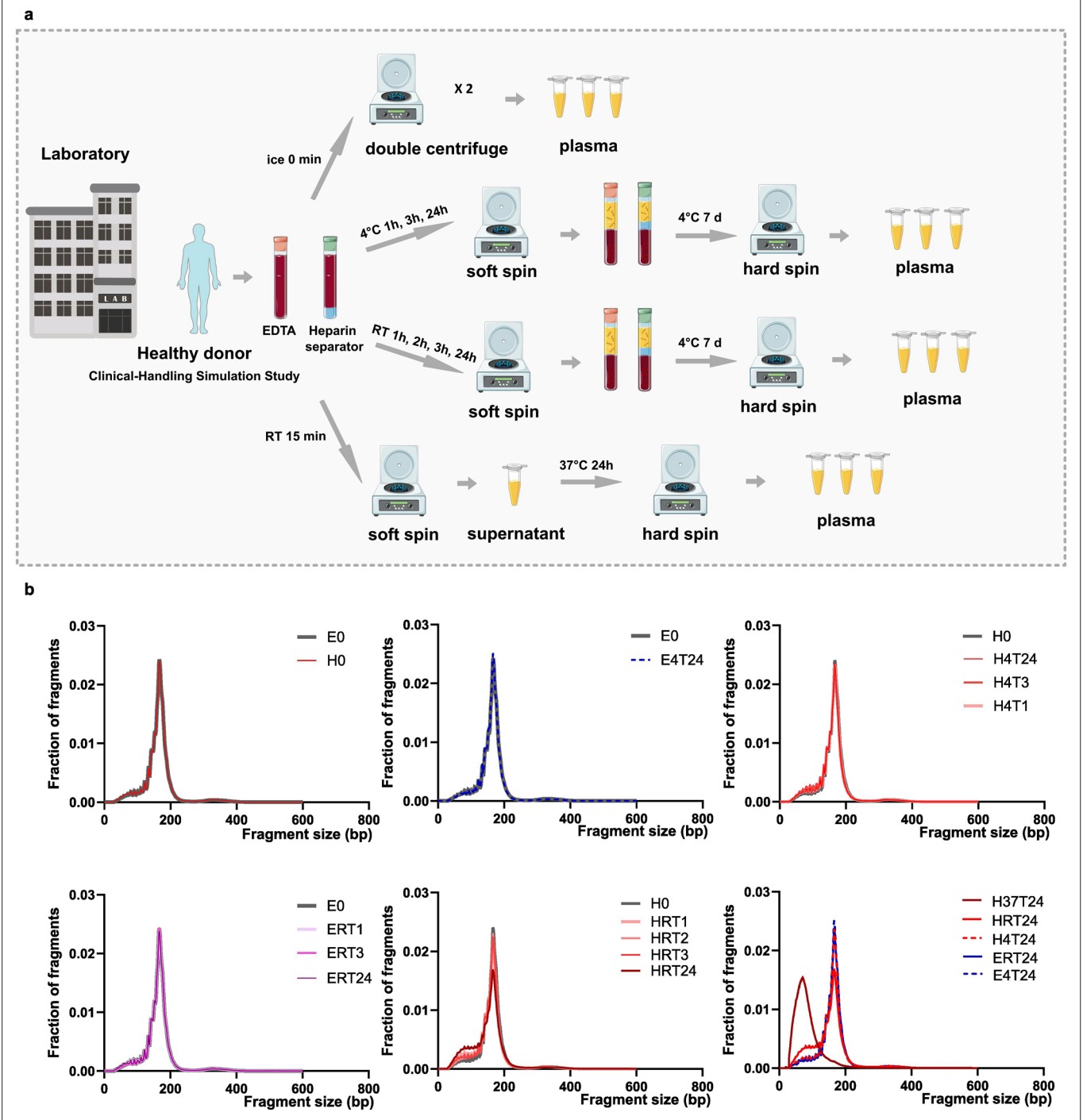

**Figure 2.** Degradation evaluation in the Clinical-Handling Simulation Study. (**a**) Schematic of the Clinical-Handling Simulation Study. Whole blood from a healthy donor was collected in EDTA tubes and heparin separator tubes and processed either immediately (ice, 0 min; double-centrifuged plasma) or after defined delays under different temperature conditions. Plasma was generated by a soft spin followed by a hard spin, including short delays at 4°C or room temperature, extended storage at 4°C for 7 days, and a stress condition at 37°C for 24 hr. (**b**) Cell-free DNA (cfDNA) fragment size distributions across all conditions, shown as fraction of fragments versus fragment size. Condition codes denote tube type and handling condition, where E indicates EDTA, H indicates heparin separator, 4 indicates 4°C, RT indicates room temperature, 37 indicates 37°C, and T indicates the incubation time in hours, with 0 indicating immediate processing.

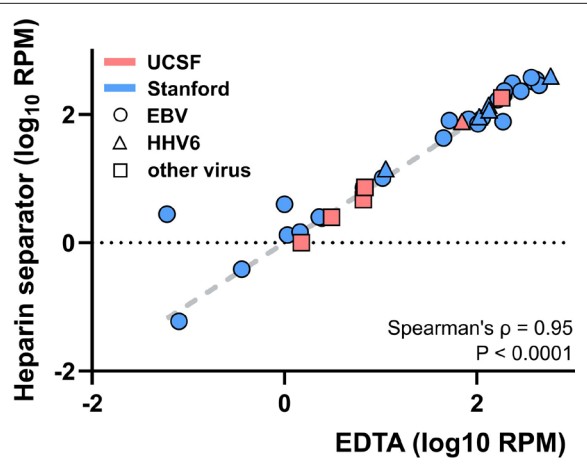

**Figure 3.** Viral load correlation of the Hospital Cohort. Correlation of plasma viral DNA reads detected by whole-genome sequencing in paired cases (n=38) collected in EDTA and heparin separators. Samples were collected at two institutions (Stanford – blue, UCSF – red, EBV – circle, HHV6 – triangle, other virus – square). EBV: Epstein-Barr virus; HHV6: human herpesvirus 6; RPM: reads per million human-aligned reads.

for the remaining CNV-positive cases are shown in *Supplementary file 3*. To summarize concordance across all paired cases, we quantified agreement using correlation of the genome-wide $\log_2$ copy ratio and visualized the distribution stratified by CNV status (*Figure 4b*). Similar results can be found in a smaller bin size of 50 kb (data uploaded to Zenodo) and a second CNV calling method, CNVkit (*Figure 4—figure supplement 1*; *Talevich et al., 2016*). CNV-positive samples (i.e. deduced tumor fraction ≥5%) exhibited consistently high concordance between tube types (n=6, Spearman's $\rho$ =0.72–0.96; *Supplementary file 3*). Tumor fraction estimates based on copy number also correlated strongly between matched samples (Spearman's $\rho$ =0.77; *Figure 4c*). These findings demonstrate that plasma from heparin separators is suitable for CNA detection and tumor fraction estimation.

## Correlated methylation profiling from heparin separator plasma

To assess the reliability of methylation profiling across blood collection tube types, we analyzed the available residual matched plasma samples in the Hospital Cohort (n=12). Using FLEXseq (*Yu et al., 2024*), a high-resolution methylation assay, we observed strong genome-wide concordance of CpG methylation levels between heparin separator and EDTA tubes (Patient 31; Spearman's $\rho$ =0.63; *Figure 5a*; other cases in *Figure 5—figure supplement 1*). To evaluate the biological relevance of the cfDNA methylation profiles, we performed cell-type deconvolution on the paired plasma (*Loyfer et al., 2023*). The results showed comparable tissue-of-origin proportions between plasma collected in heparin separator and EDTA tubes (*Figure 5b*). These findings demonstrate that cfDNA methylation signatures are well preserved in heparin separator plasma and are reliable for epigenomic applications such as cell-of-origin estimation.

## Comparable cfDNA yield deduced from the sequencing result

We quantified cfDNA concentrations (normalized to plasma volume) in the 25 matched pairs from the Tube Comparison Study in the Healthy Cohort and the Hospital Cohort with sufficient leftover DNA. Qubit measurements revealed variability across samples (0.14–43.19 ng/mL; *Figure 6a and b*) and poor correlation between EDTA and heparin separator data. However, Qubit measurements are related to both addressable (intact short fragments) and non-addressable (genomic length or broken) DNA, whereas NGS output is related to only addressable DNA.

To enable precise measurements of the addressable DNA, we spiked in lambda DNA at a constant quantity across all samples at the beginning of the library preparation and normalized human-aligned read counts to this internal control. This strategy produces a measure (human/lambda ratio) proportional to the addressable DNA content of the sample. This normalized ratio yielded strong concordance between EDTA and heparin separator samples ($R^2$=0.998; *Figure 6c*), indicating that library preparation effectively mitigated pre-analytical variability.

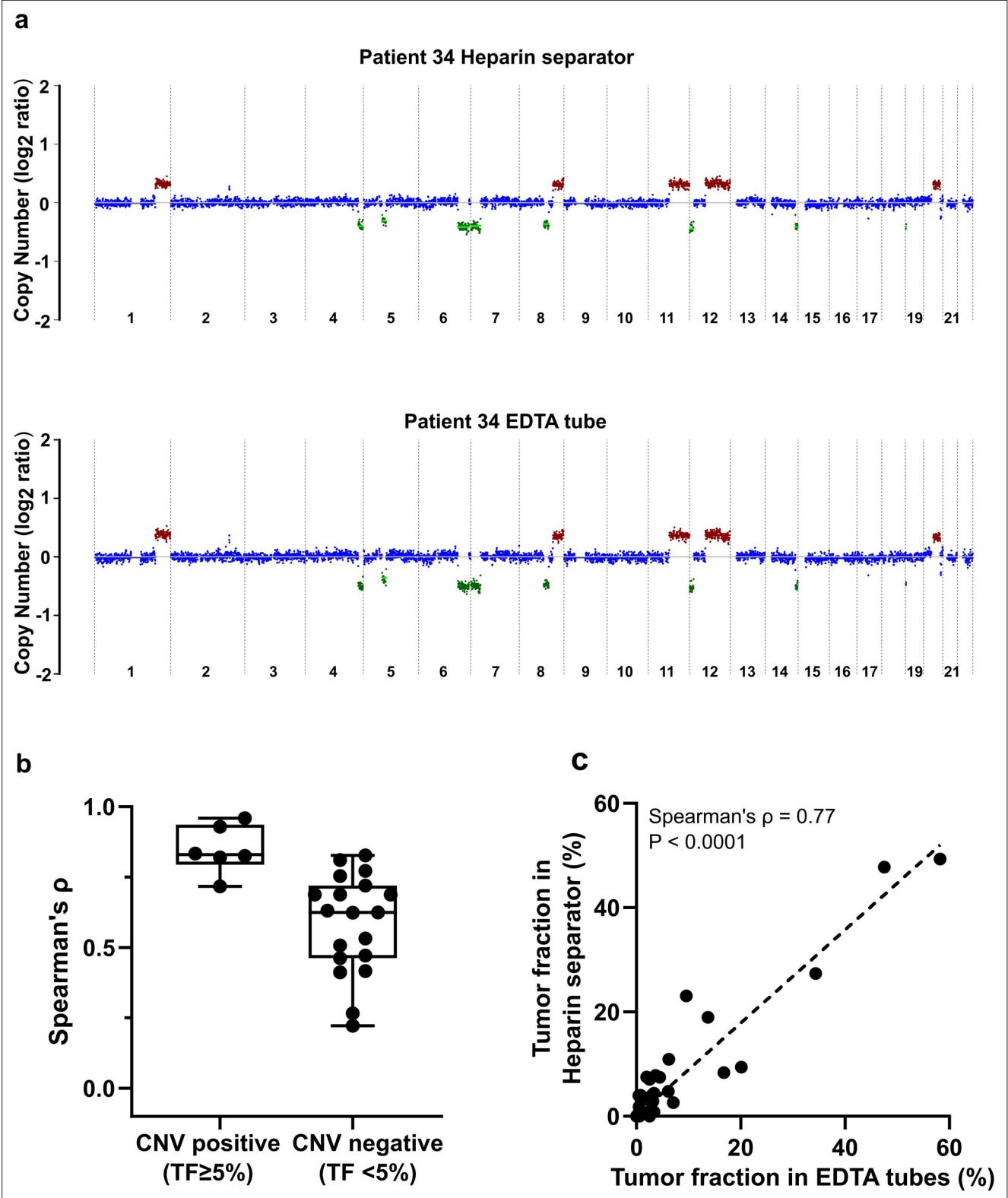

**Figure 4.** Copy number analysis of the Hospital Cohort. (**a**) An example of genome-wide copy number profiles from Patient 34. Case P171 was collected in a heparin separator; Case P187 was collected in an EDTA tube. (**b**) Boxplot of correlation of $\log_2$ copy ratio between tube types in all cases. (**c**) Comparison of tumor fraction in paired cases (n=32 pairs) inferred from copy number analysis.

The online version of this article includes the following figure supplement(s) for figure 4:

**Figure supplement 1.** CNVkit heatmap across tube types.

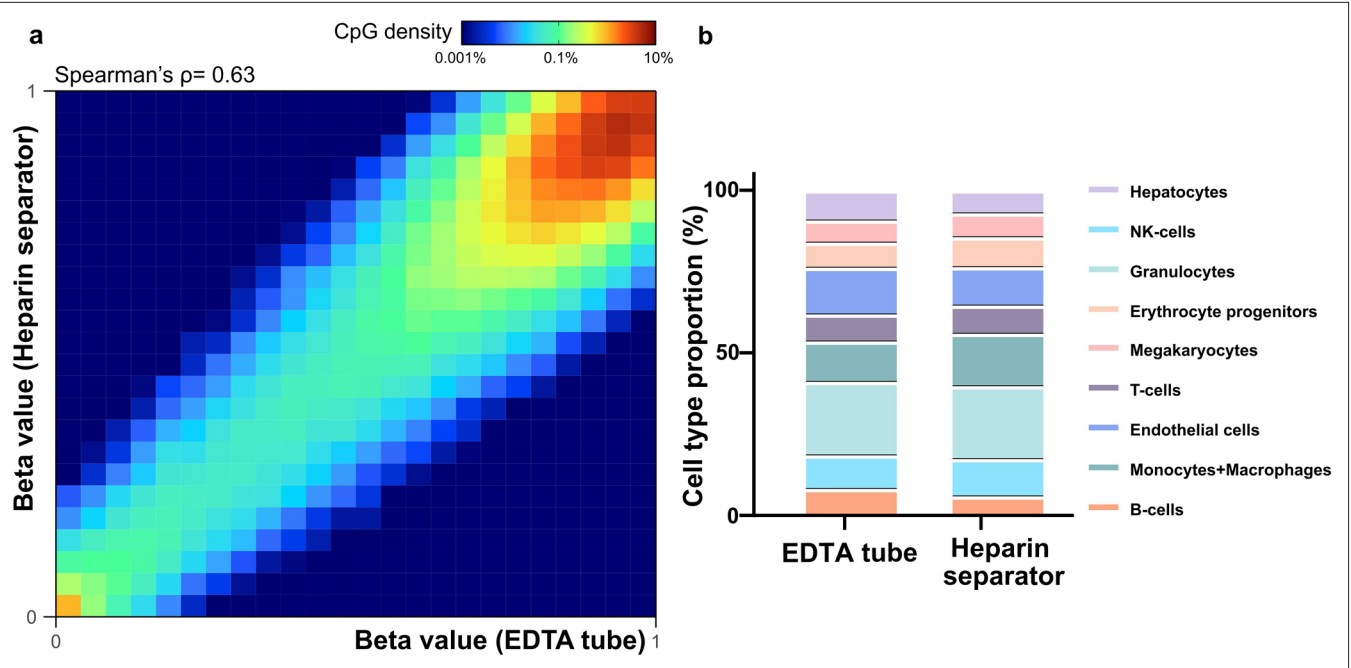

**Figure 5.** Methylation correlation and cell-type deconvolution analysis in the Hospital Cohort. (**a**) Heatmap showing methylation beta values in paired samples (Patient 31, P185 versus P168). (**b**). Estimated cell-type proportions in the paired samples (P185 versus P168).

The online version of this article includes the following figure supplement(s) for figure 5:

**Figure supplement 1.** Methylation correlation across Hospital Cohort cases.

## Leftover plasma from the heparin separator alters fragmentation profiles

Fragmentation features of cfDNA, such as size distribution and end motifs, have been linked to nuclease activity and are known to reflect host physiology and pathology (*Jiang et al., 2020*; *Ding and Lo, 2022*; *Hu et al., 2022*). To evaluate whether heparin separators preserve these fragmentation features, we examined two well-characterized fragmentomic features: size distribution and end motif. Both EDTA and heparin separator plasma samples exhibited expected peak sizes near 70 and 170 bp (*Figure 6—figure supplement 1*). However, heparin separator plasma showed a modest increase in short fragments and a slightly diminished ~166 bp peak, indicating a shorter size distribution. Analysis of 4-mer motifs at the 5′ ends of cfDNA fragments revealed significant differences in four of the six most frequent motifs (i.e. CCCA, CCAG, CCTG, and CAAA; $p<0.05$, paired Student's t-test), suggesting disrupted cleavage preference. These altered fragmentation profiles in the Hospital Cohort suggest that plasma samples collected in heparin separators may have influenced cfDNA fragmentation signatures.

## Discussion

Our study explores the untapped potential of repurposing residual plasma from heparin separators as a practical and underutilized resource for circulating cfDNA-based research and diagnostic applications. Using matched samples from the Healthy Cohort and a real-world Hospital Cohort, we demonstrate that the leftover plasma is suitable for metagenomic viral detection, copy number profiling, and genome-wide methylation analysis, with performance comparable to gold-standard collection methods.

Use of heparinized plasma has been historically limited by concerns over PCR inhibition, genomic DNA leakage, and cfDNA degradation (*Beutler et al., 1990*; *Yokota et al., 1999*; *Jung et al., 1997*; *Barra et al., 2025*; *Gerber et al., 2020*). Our study helps clarify these concerns in the context of sequencing-based assays. First, despite the well-recognized PCR-inhibitory effect of heparin, a strong concordance was observed in viral load, CNV profiles, and methylation patterns. This suggests that

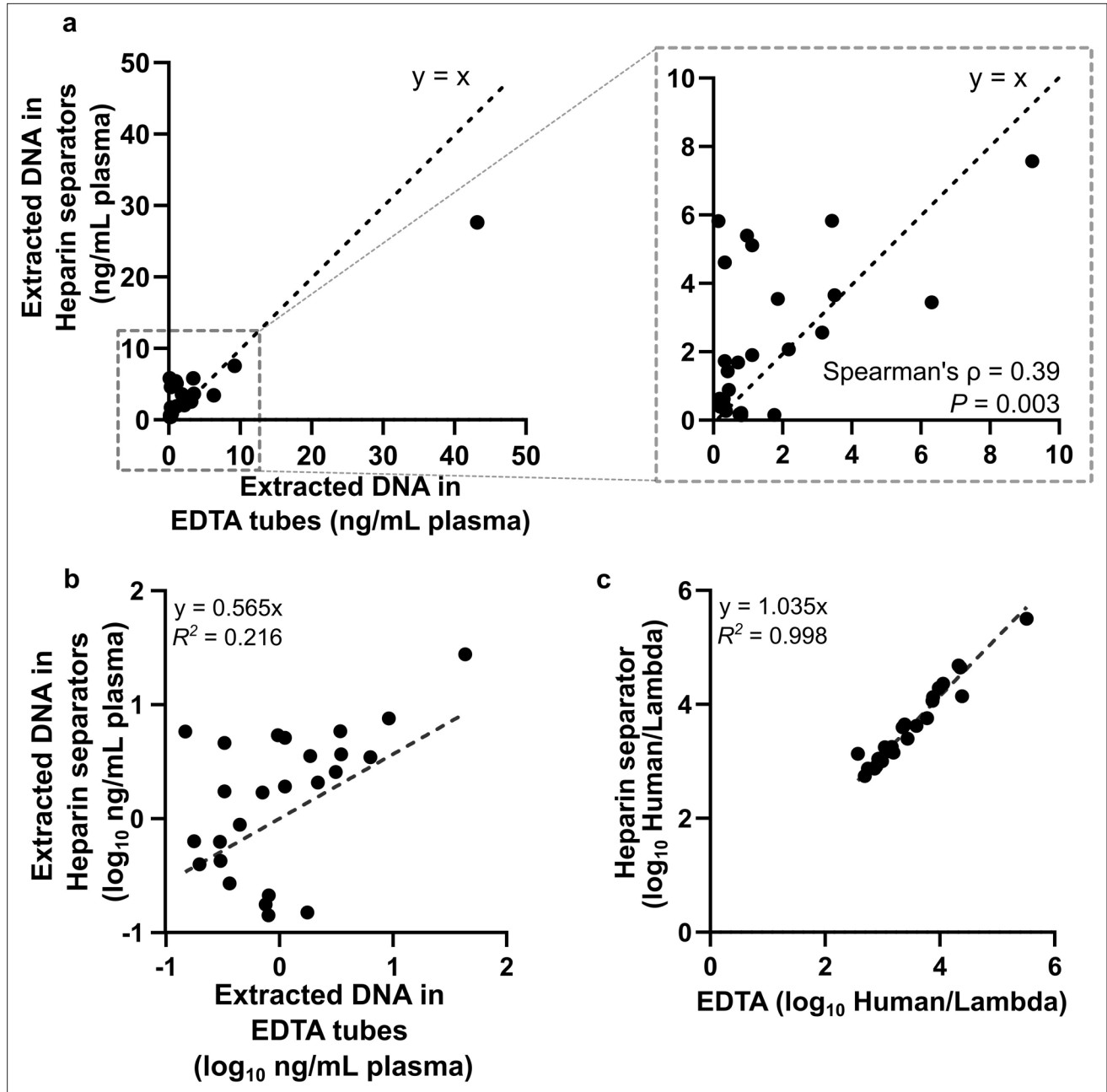

**Figure 6.** Correlation of DNA quantity. (**a**) Scatter plot of the normalized extracted DNA concentration based on Qubit reading. Data is from 20 individuals of the Hospital Cohort and 5 individuals from the Healthy Cohort using heparin separators and EDTA tubes. Normalized DNA concentration (ng/mL plasma) in heparin separator samples (y-axis) versus EDTA tube samples (x-axis), quantified by dsDNA Qubit and normalized to the input volume of plasma. (**b**) Scatter plot of the log-scale normalized concentration (Qubit readout) in the heparin separators (y-axis) versus the EDTA tube samples (x-axis). (**c**) Scatter plot of the ratio of human reads to a constant lambda phage DNA spike-in detected in the heparin separator sample (y-axis) versus the EDTA tube samples (x-axis). This spike-in measurement is proportional to the addressable DNA quantity of the sample.

The online version of this article includes the following figure supplement(s) for figure 6:

**Figure supplement 1.** Fragmentomic patterns of the Hospital Cohort.

residual heparin did not pose a practical limitation for these NGS-based assays, which, unlike PCR assays, undergo multiple rounds of DNA purification in the workflow. Second, NGS internal controls gave precise measurements of sequenceable DNA and indicated comparable amounts of sequence-able cfDNA in matched EDTA and heparin separator samples. This suggests that the confounding DNA (e.g. gDNA carryover) had minimal effect. Third, degradation risk was directly reflected by

fragmentomic analyses. Prior work showed that room temperature incubation of heparinized blood can alter cfDNA fragmentation patterns, potentially via enhanced nuclease activity (*Han et al., 2020*; *Ding et al., 2022*). However, controlled experiments mimicking clinical scenarios demonstrated that the storage temperature before the first spin is a key determinant of fragmentation integrity. Moreover, our controlled handling experiments indicate that minimizing the pre-centrifugation interval and maintaining refrigeration prior to the first spin are key to preserving cfDNA integrity in heparin separator tubes. In practice, achieving a short pre-centrifugation window is generally more feasible for inpatient collections near the core laboratory than for outpatient draws at remote clinics, which may require additional cold-chain controls.

Limitations of using residual heparin separators from chemistry tests include limited residual plasma volume (0.5–2 mL), which may not meet the input requirements of applications such as single-nucleotide variants or minimal residual disease detection. Similarly, the resultant limited sequencing depth is not optimized for high-resolution CNV detection but rather for larger-scale alterations. Moreover, variability in transport and storage conditions prior to processing, as observed in the Hospital Cohort, can affect cfDNA integrity. The observed alteration in fragmentation profiles highlights the need for cold-chain transportation protocols to minimize variability and maintain cfDNA quality. Therefore, we anticipate that this approach is optimal when specimens are from within the hospital and constrained to rapid turnaround times rather than from remote clinics. Future studies should also validate this approach in larger, clinically diverse populations and assess its integration into routine pre-analytical workflows.

In summary, residual plasma from heparin separators can serve as a viable source for cfDNA analysis, with stable results for copy number variation, viral load, and methylation profiling despite fragmentomic sensitivity to pre-analytical variability. We anticipate these specimens may be suitable for hospital biobanking, rapid and on-site testing, and scenarios where obtaining immediate informed consent in an emergent case is difficult. Repurposing residual plasma from routine blood tests could greatly simplify and expand access to cfDNA across a diverse range of patients.

## Methods
### Study design and subject selection
The Healthy Cohort consisted of two complementary experiments: (i) a Tube Comparison Study (n=5; 28–40 years of age) performed under immediate processing to assess tube-dependent effects, and (ii) a Clinical-Handling Simulation Study (n=6; 29–45 years of age) that mimics the real-world delayed processing, evaluating the effect of pre-centrifugation delays at room temperature versus 4°C. All healthy adult volunteers provided written informed consent as approved under Stanford IRB (71230). In the Tube Comparison Study, whole blood from each donor was collected via a single venipuncture into EDTA, Streck, and heparin separators. In the Clinical-Handling Simulation Study, paired blood samples were collected into EDTA and heparin separators.

The Hospital Cohort consisted of clinical samples from two institutions, Stanford and UCSF. At Stanford, 32 matched pairs of remnant samples were obtained (collected between 2022 and 2025) under a no-patient-contact protocol (IRB 58461). Patients positive for Epstein-Barr virus (EBV), human herpesvirus 6 (HHV6), BK virus, adenovirus, or Torque teno virus were identified based on routine clinical viral PCR testing performed by the Stanford or UCSF Clinical Virology Laboratories. These viral tests were typically conducted in the context of post-transplant surveillance or infectious disease diagnostics. Residual EDTA plasma samples were processed by the clinical laboratory within 4 hr of collection. Matched heparin separator plasma specimens drawn at the same time were retrieved from the Stanford Clinical Chemistry Laboratory. Only matched plasma pairs with volumes exceeding 0.5 mL were included in the study. Additional sample pairs from UCSF (n=6) were obtained (collected between 2017 and 2020) under a similar no-patient-contact IRB protocol (CA-0161925).

### Sample collection and processing
#### Healthy Cohort
In the Tube Comparison Study, samples were processed immediately after blood collection with two centrifugations (1600×*g* for 10 min and 16,000×*g* for 10 min). In the Clinical-Handling Simulation Study (*Figure 2a*), samples were incubated at either room temperature or 4°C for 0, 1, 2, 3, or 24 hr,

similar to the pre-analytical journey of a blood draw prior to the first centrifugation (1600×$g$, 10 min). Post-spin tubes were stored at 4°C for 1 week to mimic the residual plasma handling process in the hospital, followed by a second centrifugation at 16,000×$g$ for 10 min. An additional subset of heparin separator samples (n=2) was incubated at 37°C for 24 hr as a degradation control.

## Hospital Cohort

Upon arrival, plasma samples in EDTA tubes and heparin separators were immediately centrifuged at 1300×$g$ for 10 min using a Cobas 8100 automated workflow. Following completion of routine clinical testing, the residual plasma was stored and transported at approximately 4°C for up to 8 days before further processing. EDTA plasma was separated into a different container, and heparin separator plasma was kept in the same original collection tube, but physically isolated above the gel separator. These procedures were conducted by the hospital staff as part of routine clinical care in accordance with a CLIA-compliant standard operating procedure. The plasma supernatant was identified by a research team, de-identified, transferred to our research lab at refrigeration, and subjected to a second centrifugation at 16,000×$g$ for 10 min. Plasma aliquots from both EDTA tubes and heparin separators were stored at –80°C until downstream analysis. UCSF cases were processed as previously described (*Gu et al., 2021*).

## DNA extraction

cfDNA was extracted from 0.5 to 1 mL of plasma using the Maxwell RSC 48 instrument (Promega) with the Maxwell RSC ccfDNA Plasma Kit (AS1480), following the manufacturer's instructions. Samples were processed in parallel in batches. cfDNA concentrations were quantified using the Varioskan spectrophotometer (Thermo Fisher Scientific). UCSF cases were extracted as previously described (*Gu et al., 2021*).

## WGS and DNA library preparation

A median of 18.65 ng of extracted cfDNA (range: 3.00–51.70 ng) and 2.5 ng of fragmented Lambda DNA (NEB #N3013) as internal control was used for WGS library preparation with the NEBNext Ultra II DNA Library Prep Kit (New England Biolabs, #E7645L), following the manufacturer's protocol. PCR primers were NEBNext Multiplex Oligos for Illumina (NEB #E6444L). Adapter-ligated DNA was amplified for 20 cycles using the Q5 master mix, following the manufacturer's instructions. UCSF cases were processed as previously described (*Gu et al., 2021*).

## FLEXseq DNA library preparation

FLEXseq DNA libraries were prepared as previously described. Briefly, extracted cfDNA molecules were adapter-ligated, dephosphorylated, digested with MspI (NEB, #R0106T), adapter-ligated a second time with custom adapters, subjected to NEBNext Enzymatic Methyl-seq Conversion Module (New England Biolabs, #E7125L), and PCR-amplified.

## Sequencing and alignment

Paired-end sequencing was performed on the NovaSeq X platform (Illumina) at the University of Chicago Genomics Core Facility using the 25B Reagent Kit (300 cycles). For WGS data, sequencing adapters were trimmed using Cutadapt v3.5 with the following parameters: -a AGATCGGAAGAGC -A AGATCGGAAGAGC `--minimum-length 25`. Trimmed reads were aligned to the hg38 human reference genome, as well as to EBV, HHV6, and other viral genomes, using BWA v0.7.17. PCR duplicates were removed using SAMtools.

FLEXseq data were processed following the protocol described by *Yu et al., 2024*. Briefly, reads were trimmed to remove adapters and filtered based on quality scores. High-quality reads were aligned to the hg38 genome using Bismark v0.23.0. Unconverted and duplicated reads were removed using Bismark. All single-nucleotide polymorphism (SNP)-containing regions were masked for de-identification using BEDTools.

## Downstream analyses

### Viral load analysis

The ratio of reads aligned to viral genomes (e.g. EBV, HHV6) relative to every 1 million reads aligned to the human genome (hg38) was calculated and log-transformed. Spearman correlation analysis was performed between matched tube types, with the linear regression constrained to pass through the origin (0, 0).

### CNA analysis

CNAs were analyzed using ichorCNA on autosomal chromosomes (*Adalsteinsson et al., 2017*). HMMcopy's readCounter was used to generate 500 kb bin-level Wig files from aligned BAM files, which included only reads with a mapping quality score >20. Reference files for GC content (gcWig), mappability score (mapWig), and centromere positions were obtained from the ichorCNA GitHub repository corresponding to the selected bin size. Panel-of-normal Wig files were generated separately using the cases in the Healthy Cohort in the EDTA and heparin separator groups, respectively. Tumor purity was estimated based on ichorCNA outputs.

### Methylation correlation analysis

Methylation calls were extracted using the bismark_methylation_extractor. For each CpG site, the methylation level was calculated as the ratio of methylated reads to total coverage depth. CpG sites with coverage ≥30× were retained for correlation analysis between matched samples.

### Tissue-of-origin analysis

Tissue-of-origin deconvolution was performed using cell type-specific unmethylated markers as previously described by *Loyfer et al., 2023*. The top 250 unmethylated CpG markers for each cell type were extracted from the reference dataset. Markers located on sex chromosomes or overlapping with common SNPs were excluded. Deconvolution was conducted using the uxm function based on a panel of nine reference cell types: B cells, T cells, natural killer (NK) cells, granulocytes, monocytes/macrophages, endothelial cells, erythroid progenitors, megakaryocytes, and hepatocytes.

### cfDNA fragmentomics

Fragment size distributions were generated for each sample by calculating the frequency across fragment lengths between 50 and 600 bp. End motif frequencies were also calculated for each sample. The Shapiro-Wilk test was performed separately for each motif in different tubes (EDTA, heparin separator, and Streck) to assess whether the frequency distributions followed a normal distribution. For the Healthy Cohort, differences among the three collection tubes were evaluated using the Friedman test. For the Hospital Cohort, statistical significance between tube types was assessed using the paired Student's t-test. To control the false discovery rate (FDR), multiple testing corrections were applied using the Benjamini-Hochberg procedure across all tested motifs. Significance was determined based on FDR-adjusted p-values.

## Acknowledgements

This work was funded by an NIH K08 (CA230156) grant, a Burroughs-Wellcome CAMS Award to WG. We thank the members of the Stanford Clinical Molecular Genetic Pathology, Clinical Virology, and Clinical Chemistry Laboratories, and UCSF Clinical Virology and Clinical Chemistry Laboratories for saving residual specimens. We thank Debbie Chan and Dianna Ng for their critical insights at the inception of this project and Ruben Luo for facilitating Clinical Chemistry collections at Stanford. We thank the University of Chicago Genomics Facility (RRID:SCR_019196) for NovaSeq sequencing.

# Additional information

## Competing interests

Jingru Yu: The author Jingru Yu filed a patent application related to FLEXseq (PCT/US2024/056111). Lauren Ahmann: The author Lauren Ahmann filed a patent application related to FLEXseq (PCT/US2024/056111). Yvette Yao: The author Yvette Yao filed a patent application related to FLEXseq (PCT/US2024/056111). Wei Gu: The author Wei Gu filed a patent application related to FLEXseq (PCT/US2024/056111). The other authors declare that no competing interests exist.

## Funding

| Funder | Grant reference number | Author |
|---|---|---|
| NIH Office of the Director | CA230156 | Wei Gu |
| Burroughs Wellcome Fund | CAMS | Wei Gu |

The funders had no role in study design, data collection and interpretation, or the decision to submit the work for publication. For the purpose of Open Access, the authors have applied a CC BY public copyright license to any Author Accepted Manuscript version arising from this submission.

## Author contributions

Spencer C Ding, Conceptualization, Data curation, Formal analysis, Validation, Investigation, Visualization, Methodology, Writing – original draft, Project administration, Writing – review and editing; Jingru Yu, Conceptualization, Data curation, Software, Formal analysis, Visualization, Writing – review and editing; Tiepeng Liao, Data curation, Visualization, Methodology, Writing – review and editing; Lauren Ahmann, Conceptualization, Data curation, Methodology, Project administration, Writing – review and editing; Yvette Yao, Conceptualization, Data curation, Methodology, Writing – review and editing; Chandler Ho, Linlin Wang, Benjamin A Pinsky, Resources, Writing – review and editing; Wei Gu, Conceptualization, Resources, Data curation, Software, Supervision, Funding acquisition, Methodology, Writing – original draft, Project administration, Writing – review and editing

## Author ORCIDs

Spencer C Ding ⓘD https://orcid.org/0000-0002-7244-5995
Tiepeng Liao ⓘD https://orcid.org/0000-0003-0231-4401
Lauren Ahmann ⓘD https://orcid.org/0009-0008-8933-7502
Benjamin A Pinsky ⓘD https://orcid.org/0000-0001-8751-4810
Wei Gu ⓘD https://orcid.org/0000-0002-2480-6645

## Ethics

All healthy adult volunteers provided written informed consent as approved under Stanford IRB (71230). The Hospital Cohort were recruited under a no-patient-contact protocol (IRB 58461).

Reviewer #1 (Public review): https://doi.org/10.7554/eLife.108708.4.sa1
Reviewer #2 (Public review): https://doi.org/10.7554/eLife.108708.4.sa2
Author response https://doi.org/10.7554/eLife.108708.4.sa3

# Additional files

## Supplementary files

Supplementary file 1. Sample list of all cases included in this study.

Supplementary file 2. Delayed-processing conditions in the clinical-handling simulation.

Supplementary file 3. Copy number analysis across Hospital Cohort cases.

MDAR checklist

## Data availability

Raw WGS and genome-wide enriched methylation sequencing data are available for the Healthy Cohort (n = 11) with informed consent. FASTQ files were uploaded to the Sequence Read Archive (SRA), with the Bioproject ID of PRJNA1260066. Viral reads detailed statistic files (n=32), methylation calls (n=17), copy number plots (n=28), and fragment length files (n=34) are available on Zenodo (https://doi.org/10.5281/zenodo.15367093).

The following datasets were generated:

| Author(s) | Year | Dataset title | Dataset URL | Database and Identifier |
| --- | --- | --- | --- | --- |
| Gu W | 2025 | Adapting Clinical Chemistry Plasma for cfDNA liquid biopsies | https://www.ncbi.nlm.nih.gov/bioproject/?term=PRJNA1260066 | NCBI BioProject, PRJNA1260066 |
| Ding SC, Yu J, Liao T, Ahmann LS, Yao YY, Ho C, Wang L, Pinsky BA, Gu W | 2026 | Adapting Clinical Chemistry Plasma as a Source for Liquid Biopsies | https://doi.org/10.5281/zenodo.15367093 | Zenodo, 10.5281/zenodo.15367093 |

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
