## [Editor Report · eLife Assessment]

This **important** work provides a new method to extract cfDNA from residual plasma from heparin separators for molecular testing. The evidence supporting the authors' claims is **convincing**, although some further metrics should also be evaluated. This finding will be interesting to people working in epigenomics and infectious disease diagnostics.

---

## [Referee Report · Reviewer #1 (Public review)]

[Editors' note: this version has been assessed by the Reviewing Editor without further input from the original reviewers. The authors have addressed the comments raised in the previous round of review.]

Summary:

The manuscript "Adapting Clinical Chemistry Plasma as a Source for Liquid Biopsies" addresses a timely and practical question: whether residual plasma from heparin separator tubes can serve as a source of cfDNA for molecular profiling. This idea is attractive, since such samples are routinely generated in clinical chemistry labs and would represent a vast and accessible resource for liquid biopsy applications. The preliminary results are encouraging, and likely to benefit the research community.

Comments on previous revisions:

The concerns raised have been addressed. The heparin separator-based cfDNA method described in this study is likely to benefit the research community. I have no further scientific concerns.

---

## [Referee Report · Reviewer #2 (Public review)]

Summary:

The authors propose that leftover heparin plasma can serve as a source for cfDNA extraction, which could then be used for downstream genomic analyses such as methylation profiling, CNV detection, metagenomics, and fragmentomics. While the study is potentially of interest, several major limitations reduce its impact; for example, the study does not adequately address key methodological concerns, particularly cfDNA degradation, sequencing depth limitations, statistical rigor, and the breadth of relevant applications.

Strengths:

The paper provides a cheap method to extract cfDNA, which has broad application if the method is solid.

---

## [Author Response]

The following is the authors’ response to the previous reviews

**Public Reviews:**

**Reviewer #1 (Public review):**
Summary:The manuscript "Adapting Clinical Chemistry Plasma as a Source for Liquid Biopsies" addresses a timely and practical question: whether residual plasma from heparin separator tubes can serve as a source of cfDNA for molecular profiling. This idea is attractive, since such samples are routinely generated in clinical chemistry labs and would represent a vast and accessible resource for liquid biopsy applications. The preliminary results are encouraging, and likely to benefit the research community.Comments on revisions:The concerns raised have been addressed. The heparin separator-based cfDNA method described in this study is likely to benefit the research community. I have no further scientific concerns.

We appreciate the encouragement and recognition.

**Reviewer #2 (Public review):**
Summary:The authors propose that leftover heparin plasma can serve as a source for cfDNA extraction, which could then be used for downstream genomic analyses such as methylation profiling, CNV detection, metagenomics, and fragmentomics. While the study is potentially of interest, several major limitations reduce its impact; for example, the study does not adequately address key methodological concerns, particularly cfDNA degradation, sequencing depth limitations, statistical rigor, and the breadth of relevant applications.Strengths:The paper provides a cheap method to extract cfDNA, which has broad application if the method is solid.Weaknesses:(1) The introduction lacks a sufficient review of prior work. The authors do not adequately summarize existing studies on cfDNA extraction, particularly those comparing heparin plasma and EDTA plasma. This omission weakens the rationale for their study and overlooks important context.(2) The evaluation of cfDNA degradation from heparin plasma is incomplete. The authors did not compare cfDNA integrity with that extracted from EDTA plasma under realistic sample handling conditions. Their analysis (lines 90-93) focuses only on immediate extraction, which is not representative of clinical workflows where delays are common. This is in direct conflict with findings from Barra et al. (2025, LabMed), who showed that cfDNA from heparin plasma is substantially more degraded than that from EDTA plasma. A systematic comparison of cfDNA yields and fragment sizes under delayed extraction conditions would be necessary to validate the feasibility of their proposed approach.(3) The comparison of methylation profiles suffers from the same limitation. The authors do not account for cfDNA degradation and the resulting reduced input material, which in turn affects sequencing depth and data quality. As shown by Barra et al., quantifying cfDNA yield and displaying these data in a figure would strengthen the analysis. Moreover, the statistical method applied is inappropriate: the authors use Pearson correlation when Spearman correlation would be more robust to outliers and thus more suitable for methylation and other genomic comparisons.(4) The CNV analysis also raises concerns. With low-coverage WGS (~5X) from heparin-derived cfDNA, only large CNVs (>100 kb) are reliably detectable. The authors used a 500 kb bin size for CNV calling, but they did not acknowledge this as a limitation. Evaluating CNV detection at multiple bin sizes (e.g., 1 kb, 10 kb, 50 kb, 100 kb, 250 kb) would provide a more complete picture. In addition, Figure 3 presents CNV results from only one sample, which risks bias. Similar bias would exist for illustrations of CNVs from other samples in the supplementary figures provided by the authors. Again, Spearman correlation should be applied in Figure 3c, where clear outliers are visible.(5) It is important to point out that depth-based CNV calling is just one of the CNV calling methods. Other CNV calling software using SNVs, pair-reads, split-reads, and coverage depth for calling CNV, such as the software Conserting, would be severely affected by the low-quality WGS data. The authors need to evaluate at least two different software with specific algorithms for CNV calling based on current WGS data.(6) The authors omit an important application of cfDNA: somatic mutation detection. Degraded cfDNA and reduced sequencing depth could substantially impact SNV calling accuracy in terms of both recall and precision. Assessing this aspect with their current dataset would provide a more comprehensive evaluation of heparin plasma-derived cfDNA for genomic analyses.Comments on revisions:As suggested previously, the Pearson correlation analysis tends to be overstated; please replace it with Spearman correlation in the whole manuscript. Currently, the authors include both of them in the abstract, method, results, and graphics, all of which are required to be updated to only use Spearman correlation results.I don't have other concerns about the manuscript.

We entirely agree and have removed all instances of Pearson correlation from the paper, including the abstract, method, results, and graphics. Only the Spearman’s correlation was used.

We appreciate your efforts and helpful comments.